# Models of human preference
# for learning reward functions

## Abstract

The utility of reinforcement learning is limited by the alignment of reward functions with the interests of human stakeholders. One promising method for alignment is to learn the reward function from human-generated preferences between pairs of trajectory segments. These human preferences are typically assumed to be informed solely by partial return, the sum of rewards along each segment. We find this assumption to be flawed and propose modeling preferences instead as arising from a different statistic: each segment's regret, a measure of a segment's deviation from optimal decision-making. Given infinitely many preferences generated according to regret, we prove that we can identify a reward function equivalent to the reward function that generated those preferences. We also prove that the previous partial return model lacks this identifiability property without preference noise that reveals rewards' relative proportions, and we empirically show that our proposed regret preference model outperforms it with finite training data in otherwise the same setting. Additionally, our proposed regret preference model better predicts real *human* preferences and also learns reward functions from these preferences that lead to policies that are better human-aligned. Overall, this work establishes that the choice of preference model is impactful, and our proposed regret preference model provides an improvement upon a core assumption of recent research.

## 1 Introduction

Improvements in reinforcement learning (RL) have led to notable recent achievements [1–6], increasing its applicability to real-world problems. Yet, like all optimization algorithms, even *perfect* RL optimization is limited by the objective it optimizes. For RL, this objective is created in large part by the reward function. Poor alignment between reward functions and the interests of human stakeholders limits the utility of RL and may even pose catastrophic risks [7, 8].

Influential recent research has focused on reward learning from preferences over pairs of fixed-length trajectory segments. Nearly all of this recent work assumes that human preferences arise probabilistically from *only* the sum of rewards over a segment, i.e., the segment's **partial return** [9–16]. That is, these works assume that people tend to prefer trajectory segments that yield greater rewards *during the segment*. However, this preference model ignores seemingly important information about the segment's desirability, including the state values of the segment's start and end states. Separately, this partial return preference model can prefer suboptimal actions with lucky outcomes, like buying a lottery ticket.

This paper proposes an alternative preference model based on the **regret** of each segment, which is equivalent to the negated sum of an optimal policy's advantage of each transition in the segment (Section 2.2).

34 Figure 1 shows an intuitive example of when these two models disagree. Other classes of domains that
35 the models will differ on are those with constant reward until the end, including competitive games like
36 chess, go, and soccer as well as tasks for which the objective is to minimize time until reaching a goal.

37 For these two preference models, we first focus the-
38 oretically on a normative analysis (Section 3)—i.e.,
39 what preference model would we *want* humans
40 to use if we could choose—proving that reward
41 learning on infinite, exhaustive preferences with
42 our proposed regret preference model identifies a
43 reward function with the same set of optimal poli-
44 cies as the reward function with which the prefer-
45 ences are generated. We also prove that the par-
46 tial return preference model is not guaranteed to
47 identify such a reward function without preference
48 noise. We follow up with a descriptive analysis of
49 how well each of these proposed models align with
50 *actual* human preferences by collecting a human-
51 labeled dataset of preferences in a rich grid world
52 domain (Section 4) and showing that the regret pref-
53 erence model better predicts these human prefer-
54 ences (Section 5). Finally, we find that the policies
55 ultimately created through the regret preference
56 model tend to outperform those from the partial
57 return model learning—both when assessed with
58 collected human preferences or when assessed with
59 synthetic preferences (Section 6).

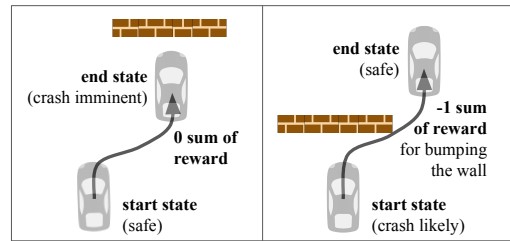

Figure 1: Two segments of a car moving at high speed
near a brick wall. Assume the right segment is opti-
mal and the left segment is suboptimal (as defined in
Sec. 2.1). The left segment has a higher sum of reward,
so the partial return preference model tends to prefer
it. The regret preference model instead tends to prefer
the right segment because optimal segments have mini-
mal regret. If we also assume deterministic transitions,
then the regret model includes the difference in values
between the start state and the end state (Eq. 3), and
the right segment would tend to be preferred because
it greatly improves its state values from start to end,
whereas the left segment's state values greatly worsen.
We suspect our human readers will also tend to prefer
the right segment.

## 2 Preference models for learning reward functions

61 We assume that the task environment is a Markov decision process (MDP) specified by the tuple $(S, A,$
62 $T, \gamma, D_0, r)$. $S$ and $A$ are the sets of possible states and actions, respectively. $T$ is a transition function,
63 $T : S \times A \to S$. $\gamma$ is the discount factor and $D_0$ is the distribution of start states. Unless otherwise
64 stated, we assume undiscounted tasks (i.e., $\gamma = 1$) that have terminal states, after which only 0 reward
65 can be received. $r$ is a reward function, $r : S \times A \times S \to \mathbb{R}$, where the reward $r_t$ at time $t$ is a function of
66 $s_t$, $a_t$, and $s_{t+1}$. An MDP$\setminus r$ is an MDP without a reward function.

67 Throughout this paper, $r$ refers to the ground-truth reward function for some MDP; $\hat{r}$ refers to a learned
68 approximation of $r$; and $\tilde{r}$ refers to any reward function (including $r$ or $\hat{r}$). A policy ($\pi : S \times A \to [0,1]$)
69 specifies the probability of an action given a state. $Q_{\tilde{r}}^*$ and $V_{\tilde{r}}^*$ refer respectively to the state-action value
70 function and state value function for an optimal policy, $\pi^*$, under $\tilde{r}$. The optimal advantage function is
71 defined as $A_{\tilde{r}}^*(s,a) \triangleq Q_{\tilde{r}}^*(s,a) - V_{\tilde{r}}^*(s)$. Throughout this paper, the ground-truth reward function $r$
72 is used to algorithmically generate preferences when they are not human-generated, is hidden during
73 reward learning, and is used to evaluate the performance of optimal policies under a learned $\hat{r}$.

### 2.1 Reward learning from pairwise preferences

75 A reward function can be learned by minimizing the cross-entropy loss—i.e., maximizing the
76 likelihood—of observed human preferences, a common approach in recent literature [9–11, 14, 16].

77 **Segments** Let $\sigma$ denote a segment starting at state $s_{\sigma,0}$. Its length $|\sigma|$ is the number of transitions within
78 the segment. A segment includes $|\sigma|+1$ states and $|\sigma|$ actions: $(s_{\sigma,0}, a_{\sigma,0}, s_{\sigma,1}, a_{\sigma,1}, ..., s_{\sigma,|\sigma|})$. In this
79 problem setting, segments lack any reward information. As shorthand, we define $\sigma_t \triangleq (s_{\sigma,t}, a_{\sigma,t}, s_{\sigma,t+1})$.
80 A segment $\sigma$ is **optimal** with respect to $\tilde{r}$ if, for every $i \in \{1,...,|\sigma|\text{-}1\}$, $Q_{\tilde{r}}^*(s_{\sigma,i}, a_{\sigma,i}) = V_{\tilde{r}}^*(s_{\sigma,i})$. A
81 segment that is not optimal is **suboptimal**. Given some $\tilde{r}$ and a segment $\sigma$, $\tilde{r}_t \triangleq \tilde{r}(s_{\sigma,t}, a_{\sigma,t}, s_{\sigma,t+1})$,
82 and the **partial return** of a segment $\sigma$ is $\sum_{t=0}^{|\sigma|-1} \gamma^t \tilde{r}_t$, denoted in shorthand as $\Sigma_\sigma r$.

**Preference datasets**  Each preference over a pair of segments creates a sample $(\sigma_1, \sigma_2, \mu)$ in a preference dataset $D_\succ$. Vector $\mu = \langle \mu_1, \mu_2 \rangle$ represents the preference; specifically, if $\sigma_1$ is preferred over $\sigma_2$, denoted $\sigma_1 \succ \sigma_2$, $\mu = \langle 1, 0 \rangle$. $\mu$ is $\langle 0, 1 \rangle$ if $\sigma_1 \prec \sigma_2$ and is $\langle 0.5, 0.5 \rangle$ for $\sigma_1 \sim \sigma_2$ (no preference).

**Loss function**  To learn a reward function from a preference dataset, $D_\succ$, a common assumption is that these preferences were generated by a preference model $P$ that arises from an unobservable *ground-truth* reward function $r$. We approximate $r$ by minimizing cross-entropy loss to learn $\hat{r}$:

$$loss(\hat{r}, D_\succ) = -\sum_{(\sigma_1, \sigma_2, \mu) \in D_\succ} \mu_1 \log P(\sigma_1 \succ \sigma_2 | \hat{r}) + \mu_2 \log P(\sigma_1 \prec \sigma_2 | \hat{r}) \tag{1}$$

This loss is under-specified until $P(\sigma_1 \succ \sigma_2 | \hat{r})$ is defined, which is the focus of this paper. We show that the common model of preference probabilities is flawed and introduce an improved preference model.

**Preference models**  A preference model determines the probability of one trajectory segment being preferred over another, $P(\sigma_1 \succ \sigma_2 | \tilde{r})$. Preference models could be applied to model preferences provided by humans or other systems. Preference models can also directly generate preferences, and in such cases we refer to them as **preference generators**.

## 2.2  Choice of preference model: partial return and regret

**Partial return**  Recent work assumes human preferences are generated by a Boltzmann distribution over the two segments' partial returns [9–16], expressed here as a logistic function[1]:

$$P_{\Sigma_r}(\sigma_1 \succ \sigma_2 | \tilde{r}) = logistic\left(\Sigma_{\sigma_1} \tilde{r} - \Sigma_{\sigma_2} \tilde{r}\right). \tag{2}$$

**Regret**  We introduce an alternative preference model based on the regret of each transition in a segment. We first focus on segments with deterministic transitions. For a transition $(s_t, a_t, s_{t+1})$ in a deterministic segment, $regret_d(\sigma_t | \tilde{r}) \triangleq V_{\tilde{r}}^*(s_{\sigma,t}) - [\tilde{r}_t + V_{\tilde{r}}^*(s_{\sigma,t+1})]$. For a full deterministic segment,

$$regret_d(\sigma | \tilde{r}) \triangleq \sum_{t=0}^{|\sigma|-1} regret_d(\sigma_t | \tilde{r}) = V_{\tilde{r}}^*(s_{\sigma,0}) - (\Sigma_\sigma \tilde{r} + V_{\tilde{r}}^*(s_{\sigma,|\sigma|})), \tag{3}$$

with the right-hand expression arising from cancelling out intermediate state values. Therefore, deterministic regret measures how much the segment reduces expected return from $V_{\tilde{r}}^*(s_{\sigma,0})$. An optimal segment, $\sigma^*$, always has 0 regret, and a suboptimal segment, $\sigma^{\neg *}$, will always have positive regret, a intuitively appealing property that also plays a role in the identifiability proof of Theorem 3.1.

Stochastic transitions, however, can result in $regret_d(\sigma^* | \hat{r}) > regret_d(\sigma^{\neg *} | \tilde{r})$, losing the property above. To retain it, we note that the effect on expected return of transition stochasticity from a transition $(s_t, a_t, s_{t+1})$ is $[\tilde{r}_t + V_{\tilde{r}}^*(s_{t+1})] - Q_{\tilde{r}}^*(s_t, a_t)$ and add this expression once per transition to get $regret(\sigma)$, removing the subscript $d$ that refers to determinism. The regret for a single transition becomes $regret(\sigma_t | \tilde{r}) = [V_{\tilde{r}}^*(s_{\sigma,t}) - [\tilde{r}_t + V_{\tilde{r}}^*(s_{\sigma,t+1})]] + [[\tilde{r}_t + V_{\tilde{r}}^*(s_{\sigma,t+1})] - Q_{\tilde{r}}^*(s_{\sigma,t}, a_{\sigma,t})] = V_{\tilde{r}}^*(s_{\sigma,t}) - Q_{\tilde{r}}^*(s_{\sigma,t}, a_{\sigma,t}) = -A_{\tilde{r}}^*(s_{\sigma,t}, a_{\sigma,t})$. Regret for a full segment is

$$regret(\sigma | \tilde{r}) = \sum_{t=0}^{|\sigma|-1} regret(\sigma_t | \tilde{r}) = \sum_{t=0}^{|\sigma|-1} \left[ V_{\tilde{r}}^*(s_{\sigma,t}) - Q_{\tilde{r}}^*(s_{\sigma,t}, a_{\sigma,t}) \right] = \sum_{t=0}^{|\sigma|-1} -A_{\tilde{r}}^*(s_{\sigma,t}, a_{\sigma,t}). \tag{4}$$

The regret preference model is the Boltzmann distribution over negated regret:

$$P_{regret}(\sigma_1 \succ \sigma_2 | \tilde{r}) \triangleq logistic\left(regret(\sigma_2 | \tilde{r}) - regret(\sigma_1 | \tilde{r})\right). \tag{5}$$

Lastly, we note that if two segments have deterministic transitions, end in terminal states, and have the same starting state, this regret model reduces to the partial return model: $P_{regret}(\cdot | \tilde{r}) = P_{\Sigma_r}(\cdot | \tilde{r})$.

**Algorithms in this paper**  All algorithms in the body of this paper are defined simply as "minimize Equation 1". They differ only in how the preference probabilities are calculated. All reward function learning via partial return uses Equation 2. We use two algorithms for reward function learning

---

[1]See Appendix B for a derivation of this logistic expression from a Boltzmann distribution with a temperature of 1. Unless otherwise stated, we ignore the temperature because scaling reward has the same effect.

118   via regret. The theory in Section 3 assumes exact measurement of regret, using Equation 5. Our
119   experimental results in Section 6 use Equation 6 to approximate regret. Appendix B introduces other
120   algorithms that use Equation 1, as well as one in Appendix B.2 that generalizes Equation 1.

121   **Regret as a model for human preference**   $P_{regret}$ makes at least three assumptions worth noting.
122   First, it keeps the assumption that human preferences follow a Boltzmann distribution over some
123   statistic, which is a common model of choice behavior in economics and psychology, where it is
124   called the Luce-Shepard choice rule [17, 18]. Second, $P_{regret}$ implicitly assumes humans can identify
125   optimal and suboptimal segments when they see them, which will less true in domains where the human
126   has less expertise. Lastly, $P_{regret}$ assumes that in stochastic settings where the best *outcome* may only
127   result from suboptimal decisions (e.g., buying a lottery ticket), humans instead prefer optimal *decisions*.
128   We suspect humans are capable of expressing either type of preference—based on decision quality
129   or desirability of outcomes—and can be influenced by training or the preference elicitation interface.
130   In practice we determine that the regret model produces improvements over the partial-return model
131   (Section 6), and its assumptions represent an opportunity for follow-up research.

132   **Alternative methods for learning reward functions**   Other methods for learning reward functions
133   include inverse reinforcement learning from demonstrations [19, 20] (discussed in Appendix B.5) and
134   inverse reward design from trial-and-error reward design in multiple instances of a task domain [21].

## 3   Theoretical comparisons

136   In this section, we consider how different ways of generating preferences affect reward inference, setting
137   aside whether humans can be influenced to give preferences in accordance with a specific preference
138   method. In economic terms, this analysis—and all of our analyses with synthetic preferences—could
139   be considered a normative analysis. In artificial intelligence, this analysis might be cast as a step
140   towards defining criteria for a rational preference model.

141   **Definition 3.1** (An identifiable preference model). *For a preference model $P$, assume an infinite*
142   *dataset $D_{\succ}$ of $n$-length pairs of segments is constructed by repeatedly choosing $(\sigma_1, \sigma_2)$ and sampling*
143   *a label $\mu \sim P(\sigma_1 \succ \sigma_2 | r)$, using $P$ as a preference generator. Further assume that in this dataset, all*
144   *possible $n$-length segment pairs appear infinitely many times. For some MDP\r $M$, let $M_{\tilde{r}}$ be $M$ with*
145   *the reward function $\tilde{r}$. Let $\Pi^*_{\tilde{r}}$ be the set of optimal policies in $M_{\tilde{r}}$. Let reward-equivalence class $\Re$ be*
146   *the set of all reward functions such that if $r_1, r_2 \in \Re$ then $\Pi^*_{r_1} = \Pi^*_{r_2}$. Preference model $P$ is **identifiable***
147   *if, for any choice of $n$ and $M_r$, any $\hat{r} = argmin_{\tilde{r}, D_{\succ}}[loss(\tilde{r})]$—for the cross-entropy loss (Eqn. 1),*
148   *with $P$ as the preference model—is in the same reward equivalence class as $r$. I.e., $\Pi^*_r = \Pi^*_{\hat{r}}$.*

149   **Theorem 3.1** ($P_{regret}$ is identifiable). *Let $P_{regret}$ be any function such that if $regret(\sigma_1 | \tilde{r}) <$*
150   *$regret(\sigma_2 | \tilde{r})$, $P_{regret}(\sigma_1 \succ \sigma_2 | \tilde{r}) > 0.5$, and if $regret(\sigma_1 | \tilde{r}) = regret(\sigma_2 | \tilde{r})$, $P_{regret}(\sigma_1 \succ \sigma_2 | \tilde{r}) =$*
151   *$0.5$. $P_{regret}$ is identifiable.*

152   This class of regret preference models includes but is not limited to the Boltzmann distribution of Eqn. 5
153   and the narrower class that Theorem 3.1 focuses upon.

154   **Theorem 3.2** (Noiseless $P_{\Sigma_r}$ is not identifiable). *Let $P_{\Sigma_r}$ be any function such that if $\Sigma_{\sigma_1} \tilde{r} > \Sigma_{\sigma_2} \tilde{r}$,*
155   *$P_{\Sigma_r}(\sigma_1 \succ \sigma_2 | \tilde{r}) = 1$, and if $\Sigma_{\sigma_1} \tilde{r} = \Sigma_{\sigma_2} \tilde{r}$, $P_{\Sigma_r}(\sigma_1 \succ \sigma_2 | \tilde{r}) = 0.5$. There exists an MDP in which $P_{\Sigma_r}$ is*
156   *not identifiable.*

157   Appendix C contains a proof of Theorem 3.1 and two proofs by example for Theorem 3.2, each
158   focusing on a different weakness of $P_{\Sigma_r}$. The first proof by example reveals issues when learning
159   reward functions with stochastic transitions with either $P_{\Sigma_r}$ or *deterministic* $P_{regret_d}$. These issues
160   directly correspond to the need for preferences over distributions over outcomes (i.e., lotteries) to
161   construct a cardinal utility function (see Russell and Norvig [22, Ch. 16]). Note that the noiseless
162   version of $P_{\Sigma_r}$ in Theorem 3.2 is achieved in the limit as reward values are scaled higher; equivalently,
163   one could include a Boltzmann temperature parameter in Equation 2 and scale it towards 0. Intuitively,
164   Theorem 3.2 says that $P_{\Sigma_r}$ is not identifiable without the distribution over preferences providing
165   information about the proportions of rewards with respect to each other. In contrast, to be identifiable,
166   the regret preference model does not require this preference error (though it can presumably benefit
167   from it in certain contexts).

# 4 Creating a human-labeled preference dataset

To empirically investigate the consequences of each preference model when learning reward from *human* preferences, we created a preference dataset labeled by human subjects via Amazon Mechanical Turk. This data collection was IRB-approved. Appendix D adds detail to the content below.

## 4.1 The general delivery domain

The delivery domain consists of a grid of cells, each of a specific road surface type. The delivery agent's state is its location. The agent's action space is moving one cell in one of the four cardinal directions. The episode can terminate either at the destination for $+50$ reward or in failure at a sheep for $-50$ reward. The reward for a non-terminal transition is the sum of any reward components. Cells with a white road surface have a $-1$ reward component, and cells with brick surface have a $-2$ component. Additionally, each cell may contain a coin $(+1)$ or a roadblock $(-1)$. Coins do not disappear and at best cancel out the road surface cost. Actions that would move the agent into a house or beyond the grid's perimeter result in no motion and receive reward that includes the current cell's surface reward component but not any coin or roadblock components. In this work, the start state distribution, $D_0$, is always uniformly random over non-terminal states. This domain was designed to permit subjects to easily identify bad behavior yet also to be difficult for them to determine *optimal* behavior from most states, which is representative of many common tasks.

### 4.1.1 The delivery task

We chose one instantiation of the delivery domain for gathering our dataset of human preferences. This specific MDP has a $10 \times 10$ grid. From every state, the highest return possible involves reaching the goal, rather than hitting a sheep or perpetually avoiding termination. Figure 2 shows this task.

## 4.2 The user interface and survey

This subsection describes the three main stages of the experimental session. A video showing the full experimental protocol can be seen at bit.ly/humanprefs.

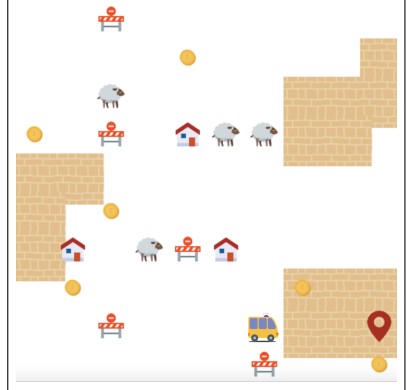

Figure 2: The delivery task used to gather human preferences. The yellow van is the agent and the red inverted teardrop is the destination.

**Teaching subjects about the task** Subjects first view instructions describing the general domain. To avoid the jargon of "return" and "reward," these terms are mapped to equivalent values in US dollars, and the instructions describe the goal of the task as maximizing the delivery vehicle's financial outcome, where the reward components are specific financial impacts. This information is shared amongst interspersed interactive episodes, in which the subject controls the agent in domain maps that are each designed to teach one or two concepts. Our intention during this stage is to inform the later preferences of the subject by teaching them about the domain's dynamics and its reward function, as well as to develop the subject's sense of how desirable various behaviors are. At the end of this stage, the subject controls the agent for two episodes in the specific delivery task shown in Figure 2.

**Preference elicitation** After each subject is trained to understand the task, they indicate their preferences between 40–50 randomly-ordered pairs of segments, using the interface shown in Figure 3. The users select a preference, no preference ("same"), or "can't tell". In this work, we exclude responses labeled "can't tell", though one might alternatively try to extract information from these responses.

**Users' task comprehension** Subjects then answered questions testing their understanding of the task, and we removed their data if they scored poorly. We also removed a subject's data if they preferred colliding the vehicle into a sheep over not doing so, which we interpreted as poor task understanding or inattentiveness. This filtered dataset contains 1812 preferences from 50 subjects.

### 4.3 Selection of segment pairs for labeling

We collected human preferences in two stages, each with different methods for selecting which segment pairs to present for labeling. The second stage's sole purpose was to improve the reward-learning performance of $P_{\Sigma_r}$. Without second-stage data, $P_{\Sigma_r}$ compared even worse to $P_{regret}$ than in the results described in Section 6 (see Appendix **??**). Both stages'

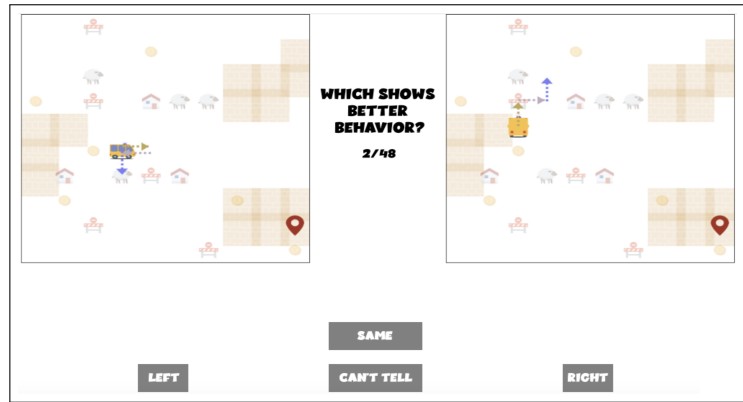

Figure 3: Interface shown to subjects during preference elicitation.

data are combined and used as a single dataset. These methods and their justification are described in Appendix D.3.

## 5 Descriptive results

This section considers how well different preference models explain our dataset of human preferences.

### 5.1 Correlations between preferences and segment statistics

We hypothesize that the values of segments' start and end states—which are included in $P_{regret}$ but not in $P_\Sigma$—affect human preferences, independent of partial return. To simplify analysis, we combine the two parts of $regret_d(\sigma|r)$ that are additional to $\Sigma_\sigma \tilde{r}$ and introduce the following shorthand: $\Delta_\sigma V_{\tilde{r}} \triangleq V_{\tilde{r}}^*(s_{\sigma,|\sigma|}) - V_{\tilde{r}}^*(s_{\sigma,0})$. Note that with an algebraic manipulation (see Appendix E.1), $regret_d(\sigma_2|\tilde{r}) - regret_d(\sigma_1|\tilde{r}) = (\Delta_{\sigma_1} V_{\tilde{r}} - \Delta_{\sigma_2} V_{\tilde{r}}) + (\Sigma_{\sigma_1}\tilde{r} - \Sigma_{\sigma_2}\tilde{r})$. Therefore, on the diagonal line in Figure 4, $regret_d(\sigma_2|r) = regret_d(\sigma_1|r)$, making the $P_{regret_d}$ preference model indifferent.

The dataset of preferences is visualized in Figure 4. This plot shows how $\Delta_\sigma V_r$ has influence independent of partial return by focusing only on points at a chosen $y$-axis value; if the colors along the corresponding horizontal line reddens as the $x$-axis value increases, then $\Delta_\sigma V_r$ appears to have independent influence. To statistically test for independent influence of $\Delta_\sigma V_r$ on preferences, we consider subsets of data where $\Sigma_{\sigma_1} r - \Sigma_{\sigma_2} r$ is constant. For $\Sigma_{\sigma_1} r - \Sigma_{\sigma_2} r = -1$ and $\Sigma_{\sigma_1} r - \Sigma_{\sigma_2} r = -2$, the only values with more than 30 samples that also include informative samples with both negative and positive values of $regret(\sigma_1|r) - regret(\sigma_2|r)$, the Spearman's rank correlations between $\Delta_\sigma V_r$ and the preferences are significant ($r >= 0.3$, $p < 0.0001$). This result indicates that $\Delta_\sigma V_r$ *influences human preferences independent of partial return*, validating our hypothesis that humans form preferences based on information about segments' start states and end states, not only partial returns.

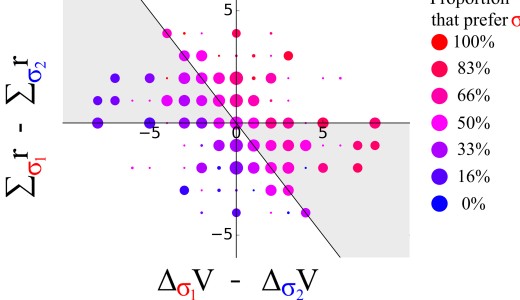

Figure 4: Proportions at which subjects preferred each segment in a pair, plotted by the difference in the segments' changes in state values (x-axis) and partial returns (y-axis). The diagonal line shows points of preference indifference for $P_{regret}$. Points of indifference for $P_\Sigma$ lie on the x-axis. The shaded gray area indicates where the two models disagree, each giving a different segment a preference probability greater than 0.5. Each circle's area is proportional to the number of samples it describes.

| Preference model | Loss |
|---|---|
| $P(\cdot) = 0.5$ (uninformed) | 0.69 |
| $P_{\Sigma_r}$ (partial return) | 0.62 |
| $P_{regret}$ | **0.57** |

Table 1: Mean cross-entropy test loss over 10-fold cross validation (n=1812) from predicting human preferences. Lower is better.

 **5.2 Likelihood of human preferences under different preference models**

263 To examine how well each preference model predicts human preferences, we calculate the cross-
264 entropy loss for each model (Eqn. 1)—i.e., the negative log likelihood—of the preferences in our
265 dataset. Scaling reward by a constant factor does not affect the set of optimal policies. Therefore,
266 throughout this work we ensure that our analyses of preference models are insensitive to reward scaling.
267 To do so for this specific analysis, we conduct 10-fold cross validation to learn a reward scaling factor
268 for each of $P_{regret}$ and $P_{\Sigma_r}$. Table 1 shows that the loss of $P_{regret}$ is lower than that of $P_{\Sigma_r}$, indicating
269 that it is more reflective of how people actually express preferences.

270 # 6   Results from learning reward functions

271 Analysis of a preference model's predictions of human preferences is informative, but such predictions
272 are a means to the ends of learning human-aligned reward functions and policies. We now examine each
273 preference model's performance on these ends. In all cases, we learn a reward function $\hat{r}$ according
274 to Eqn. 1 and apply value iteration [23] to find the approximately optimal $Q^*_{\hat{r}}$ function. For this $Q^*_{\hat{r}}$,
275 we then evaluate the mean return of the maximum-entropy optimal policy—which chooses uniformly
276 randomly among all *optimal* actions—with respect to the ground-truth reward function $r$, over $D_0$.
277 To compare performance across different MDPs, the mean return of a policy $\pi$, $V^\pi_r$, is normalized
278 to $(V^\pi_r - V^U_r)/V^*_r$, where $V^*_r$ is the optimal expected return and $V^U_r$ is the expected return of the
279 uniformly random policy (both given $D_0$). Normalized mean return above 0 is better than $V^U_r$. Optimal
280 policies have a normalized mean return of 1, and we consider above 0.9 to be *near optimal*.

281 ## 6.1   An algorithm to learn reward functions with $regret(\sigma_\sigma|\hat{r})$

282 Algorithm 1 is a general algorithm for learning a *linear* reward function according to $P_{regret}$. This
283 regret-specific algorithm only changes the regret-based algorithm from Section 2.2 by replacing
284 Equation 5 with a tractable approximation of regret, avoiding expensive repeated evaluation of $V^*_{\hat{r}}(\cdot)$
285 and $Q^*_{\hat{r}}(\cdot,\cdot)$ to compute $P_{regret}(\cdot|\hat{r})$ during reward learning. Specifically, successor features for a set
286 of policies are used to approximate the optimal state values and state-action values for *any* reward
287 function.

288 **Approximating $P_{regret}$ with successor features**   Following the notation of Barreto et al. [24], assume
289 the ground-truth reward is linear with respect to a feature vector extracted by $\phi : S \times A \times S \to \mathbb{R}^d$ and
290 a weight vector $\boldsymbol{w_r} \in \mathbb{R}^d$: $r(s,a,s') = \phi(s,a,s')^\top \boldsymbol{w_r}$. During learning, $\boldsymbol{w_{\hat{r}}}$ similarly expresses $\hat{r}$ as
291 $\hat{r}(s,a,s') = \phi(s,a,s')^\top \boldsymbol{w_{\hat{r}}}$.

292 Given a policy $\pi$, the successor features for $(s,a)$ are the expectation of discounted reward features
293 from that state-action pair when following $\pi$: $\boldsymbol{\psi}^\pi_{\boldsymbol{Q}}(s,a) = E^\pi[\sum_{i=t}^{\infty}\gamma^{i-t}\phi(s_t,a_t,s_{t+1})|s_t=s,a_t=a]$.
294 Therefore, $Q^\pi_{\hat{r}}(s,a) = \boldsymbol{\psi}^\pi_{\boldsymbol{Q}}(s,a)^\top \boldsymbol{w_{\hat{r}}}$. Additionally, state-based successor features can be calculated
295 from the $\boldsymbol{\psi}^\pi_{\boldsymbol{Q}}$ above as $\boldsymbol{\psi}^\pi_{\boldsymbol{V}}(s) = \sum_{a \in A}\pi(a|s)\boldsymbol{\psi}^\pi_{\boldsymbol{Q}}(s,a)$, making $V^\pi_{\hat{r}}(s) = \boldsymbol{\psi}^\pi_{\boldsymbol{V}}(s)^\top \boldsymbol{w_{\hat{r}}}$.

296 Given a set $\Psi_Q$ of state-action successor feature functions and a set $\Psi_V$ of state successor feature func-
297 tions for various policies and given a reward function via $\boldsymbol{w_{\hat{r}}}$, $Q^{\pi^*}_{\hat{r}}(s,a) \geq max_{\boldsymbol{\psi_Q} \in \Psi_Q}[\boldsymbol{\psi}^\pi_{\boldsymbol{Q}}(s,a)^\top \boldsymbol{w_{\hat{r}}}]$
298 and $V^{\pi^*}_{\hat{r}}(s) \geq max_{\boldsymbol{\psi_V} \in \Psi_V}[\boldsymbol{\psi}^\pi_{\boldsymbol{V}}(s)^\top \boldsymbol{w_{\hat{r}}}]$ [24], so we use these two maximizations as approximations of
299 $Q^*_{\hat{r}}(s,a)$ and $V^*_{\hat{r}}(s)$, respectively. In practice, to enable gradient-based optimization with current tools,
300 the maximization in this expression is replaced with the softmax-weighted average, making the loss
301 function linear. Focusing first on the approximation of $V^*_{\hat{r}}(s)$, for each $\boldsymbol{\psi_V} \in \Psi_V$, a softmax weight is
302 calculated for $\boldsymbol{\psi}^\pi_{\boldsymbol{V}}(s)$: $softmax_{\Psi_V}(\boldsymbol{\psi}^\pi_{\boldsymbol{V}}(s)^\top \boldsymbol{w_{\hat{r}}}) \triangleq [(\boldsymbol{\psi}^\pi_{\boldsymbol{V}}(s)^\top \boldsymbol{w_{\hat{r}}})^{1/T}]/[(\sum_{\boldsymbol{\psi'_V} \in \Psi_V}\boldsymbol{\psi'}^\pi_{\boldsymbol{V}}(s)^\top \boldsymbol{w_{\hat{r}}})^{1/T}]$,
303 where temperature $T$ is a constant hyperparameter. The resulting approximation of $V^*_{\hat{r}}(s)$ is there-
304 fore defined as $\tilde{V}^*_{\hat{r}}(s) \triangleq \sum_{\boldsymbol{\psi_V} \in \Psi_V} softmax_{\Psi_V}(\boldsymbol{\psi}^\pi_{\boldsymbol{V}}(s)^\top \boldsymbol{w_{\hat{r}}})[\boldsymbol{\psi}^\pi_{\boldsymbol{V}}(s)^\top \boldsymbol{w_{\hat{r}}}]$. Similarly, to approxi-
305 mate $Q^*_{\hat{r}}(s,a)$, $softmax_{\Psi_Q}(\boldsymbol{\psi}^\pi_{\boldsymbol{Q}}(s,a)^\top \boldsymbol{w_{\hat{r}}}) \triangleq [(\boldsymbol{\psi}^\pi_{\boldsymbol{Q}}(s,a)^\top \boldsymbol{w_{\hat{r}}})^{1/T}]/[(\sum_{\boldsymbol{\psi'_Q} \in \Psi}\boldsymbol{\psi'}^\pi_{\boldsymbol{Q}}(s,a)^\top \boldsymbol{w_{\hat{r}}})^{1/T}]$
306 and $\tilde{Q}^*_{\hat{r}}(s,a) \triangleq \sum_{\boldsymbol{\psi_Q} \in \Psi_Q} softmax_{\Psi_Q}(\boldsymbol{\psi}^\pi_{\boldsymbol{Q}}(s,a)^\top \boldsymbol{w_{\hat{r}}})[\boldsymbol{\psi}^\pi_{\boldsymbol{Q}}(s,a)^\top \boldsymbol{w_{\hat{r}}}]$. Consequently, from Eqns. 4

---

**Algorithm 1** Linear reward learning with regret preference model ($P_{regret}$), using successor features

---
1: Input: a set of reward functions and a set of policies (where one set can be $\varnothing$)
2: $\Psi \leftarrow \varnothing$
3: **for** *each reward function $r_{SF}$ or policy $\pi_{SF}$ in the input sets* **do**
4:     **if** $r_{SF}$ **then** $\pi_{SF} \leftarrow$ estimate of optimal maximum-entropy policy for $r_{SF}$
5:     estimate $\psi_Q^{\pi_{SF}}$ and $\psi_V^{\pi_{SF}}$ (if not estimated already during step 4)
6:     add $\psi_Q^{\pi_{SF}}$ to $\Psi_Q$
7:     add $\psi_V^{\pi_{SF}}$ to $\Psi_V$
8: **end for**
9: **repeat**
10:     optimize $\boldsymbol{w_{\hat{r}}}$ by loss of Eqn. 1, calculating $\tilde{P}_{regret}(\sigma_1 \succ \sigma_2 | \hat{r})$ via Eqn. 6, using $\Psi_Q$ and $\Psi_V$
11: **until** *stopping criteria are met*
12: **return** $\boldsymbol{w_{\hat{r}}}$

---

and 5, the corresponding approximation $\tilde{P}_{regret}$ of the regret preference model is:

$$\tilde{P}_{regret}(\sigma_1 \succ \sigma_2 | \hat{r}) = logistic\left( \sum_{t=0}^{|\sigma_2|-1} \left[ \tilde{V}_{\hat{r}}^*(s_{\sigma_2,t}) - \tilde{Q}_{\hat{r}}^*(s_{\sigma_2,t}, a_{\sigma_2,t}) \right] - \sum_{t=0}^{|\sigma_1|-1} \left[ \tilde{V}_{\hat{r}}^*(s_{\sigma_1,t}) - \tilde{Q}_{\hat{r}}^*(s_{\sigma_1,t}, a_{\sigma_1,t}) \right] \right) \quad (6)$$

**The algorithm** In Algorithm 1, lines 9–12 describe the supervised-learning optimization using the approximation $\tilde{P}_{regret}$, and the prior lines create $\Psi_Q$ and $\Psi_V$. Specifically, given a set of reward functions, a corresponding set of policies is created (line 4), where each policy is an estimate of the maximum entropy policy for a reward function. Standard policy improvement methods can be used to create each such policy. Alternatively, some or all of the set of policies can be given as input directly, not derived from input reward functions. For each such policy $\pi_{SF}$, successor feature functions $\Psi_Q^{\pi_{SF}}$ and $\Psi_V^{\pi_{SF}}$ are estimated (line 5), which by default would be performed by a minor extension of a standard policy evaluation algorithm as detailed by Barreto et al. [24]. Note that the reward function that is ultimately learned is not restricted to be in the input set of reward functions, which is used only to create an approximation of regret.

The details of our instantiation of Algorithm 1 for the delivery domain can be found in Appendix F.1, along with guidance for extending it to reward functions that might be non-linear.

## 6.2 Results from synthetic preferences

Before considering human preferences, we first ask how each preference model performs when it is correct. In other words, we investigate empirically how well the preference model could perform if humans perfectly adhered to it. Recall that the ground-truth reward function, $r$, is used to create these preferences but is inaccessible to the reward-learning algorithms.

For these evaluations, either a stochastic or noiseless preference model acts a preference generator to create a preference dataset, and then the stochastic version of the same model is used for reward learning. For the noiseless case, the deterministic preference generator compares a segment pair's $\Sigma_\sigma r$ values for $P_{\Sigma_r}$ or their $regret(\sigma|r)$ values for $P_{regret}$. Note that through reward scaling the preference generators approach determinism in the limit, so this noiseless analysis examines minimal-entropy versions

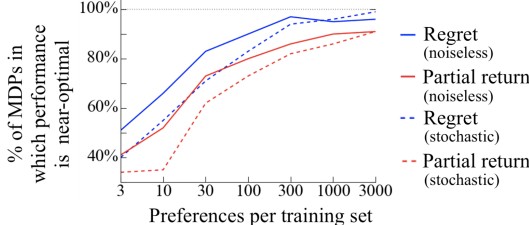

Figure 5: Performance comparison over 100 randomly generated deterministic MDPs

of the two preference-generating models. (The opposite extreme, uniformly random preferences, would remove all information from preferences and therefore is not examined.) In the stochastic case, for each preference model, each segment pair is labeled by sampling from that preference generator's output distribution (Eqs 2 or 5), using the unscaled ground-truth reward function.

We created 100 deterministic MDPs that instantiate variants of our delivery domain (see Section 4.1). To create each MDP, we sampled from sets of possible widths, heights, and reward component values, and the resultant grid cells were randomly populated with a destination, objects, and road surface types (see Appendix F.2 for details). Each segment in the preference datasets for each MDP was generated by choosing a start state and three actions, all uniformly randomly. For a set number of preferences, each method had the same set of segment pairs in its preference dataset. Figure 5 shows the percentage of MDPs in which each preference model results in near-optimal performance. The regret preference model outperforms the partial return model at every dataset size, both with and without noise. By a Wilcoxon paired signed-rank test on normalized mean returns, $p < 0.05$ for 86% of these comparisons and $p < 0.01$ for 57% of them, as reported in Appendix F.2.

Further analyses can be found in Appendix F.2, including with stochastic transitions, with different segment lengths, and while artificially lowering the discount factor (as is common in deep RL and recent work on deep reward learning from preferences).

### 6.3 Results from human preferences

We randomly assign human preferences from our gathered dataset to different numbers of same-sized partitions, resulting in different training set sizes, and test each preference model on each partition. Figure 6 shows the results. With smaller training sets (20–100 partitions), the regret preference model results in near-optimal performance more often. With larger training sets (1–10 partitions), both preference models always reach near-optimal return, but the mean return from the regret preference model is higher for all of these partitions except for 3 partitions in the 10-partition

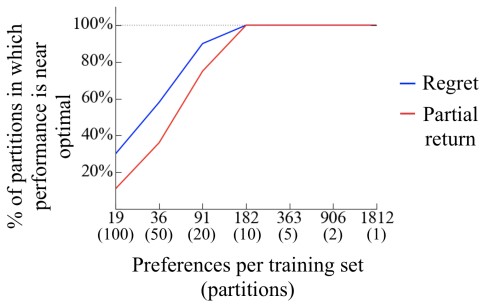

Figure 6: Performance comparison over various amounts of human preferences. Each partition has the number of preferences shown or one less.

test. Applying a Wilcoxon paired signed-rank test on normalized mean return to each group with 5 or more partitions, $p < 0.05$ for all numbers of partitions except 100 and $p < 0.01$ for 20 and 50 partitions.

## 7 Conclusion

Over numerous evaluations with human preferences, our proposed regret preference model ($P_{regret}$) shows improvements summarized below over the previous partial return preference model ($P_{\Sigma_r}$). When each preference model generates the preferences for its own infinite and exhaustive training set, we prove that $P_{regret}$ identifies the set of optimal policies, whereas $P_{\Sigma_r}$ is not guaranteed to do so without preference noise that reveals the proportions of rewards with respect to each other. With finite training data of synthetic preferences, $P_{regret}$ also empirically results in learned policies that tend to outperform those resulting from $P_{\Sigma_r}$. This superior performance of $P_{regret}$ is also seen with human preferences. In summary, our analyses suggest that regret preference models are more effective both descriptively with respect to human preferences and also normatively, as the model we want humans to follow if we had the choice.

Independent of $P_{regret}$, this paper also reveals that segments' changes in state values provide information about human preferences that is not fully provided by partial return. More generally, we show that the choice of preference model impacts the performance of learned reward functions.

This study motivates several new directions for research. Future work could address any of the limitations detailed in Appendix A.1. Specifically, future work could further test the general superiority of $P_{regret}$ or apply it to deep learning settings. Additionally, *prescriptive* methods could be developed via the user interface or elsewhere to nudge humans to conform more to $P_{regret}$ or to other normatively appealing preference models. Lastly, subsequent efforts could seek preference models that are even more effective with preferences from actual humans, now that this work has provided conclusive evidence that the choice of preference model is impactful.

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
