# OpenReview forum: "Models of human preference for learning reward functions"
_NeurIPS.cc/2022/Conference — NeurIPS 2022 Submitted_

### Official Review · Reviewer_SeBN · 2022-07-10

**Rating:** 4
**Confidence:** 4
**Soundness:** 3 good
**Presentation:** 3 good
**Contribution:** 2 fair

**Summary:**

The paper aims to learn the reward function from preferences between pairs of trajectory segments by introducing a new notion of the segment's regret that is based on the advantage function. Notable, they argue that the existing approach using the partial returns in the literature is flawed due to the lack of an identifiability property: the ability to recover the reward function underlying the preference dataset. As a result, the importance of using the segment's regret in learning the reward function from preference datasets is highlighted.


**Questions:**

Please address the weakenesses mentioned above.

**Ethics Review Area:**

["I don’t know"]

**Limitations:**

The scalability of the proposed method to a more realistic environment is not demonstrated.

**Strengths And Weaknesses:**

The main contribution of the paper is to introduce a new preference model based on the advantage function. The derivation of the segment's regret as the advantage function is sound and reasonable.

However, there are several weaknesses in the paper.

1, if I am not mistaken, the proposed regret model is limited to the case of deterministic policies. How is it extended to stochastic policies?

2, compared with existing approaches that use partial returns, computing the advantage function in the segment's regret is computationally expensive (as mentioned in Appendix A1). While the paper shows an approximation to reduce the computation in Section 6.1, it only works for the linear reward function. Hence, a more detailed discussion on the time complexity of the proposed regret vs. that of the partial returns is necessary, and the work needs to illustrate whether the proposed method is scalable to more practical problems, e.g., by including experiments with neural networks and larger state/action spaces.

3, if I am not missing anything, Theorem 3.1 and 3.2 are not comparable. While both partial returns and the segment's regret use the logistic function (Equation 2 and Equation 5), there is a difference in the two theorems: in Theorem 3.1, if $regret(\sigma1|\tilde{r}) < regret(\sigma_2|\tilde{r})$, $P_{regret}(\sigma_1 > \sigma_2|\tilde{r}) > 0.5$, in Theorem 3.2, if $\Sigma_{\sigma_1} \tilde{r} > \Sigma_{\sigma_2} \tilde{r}$, $P_{\Sigma_r}(\sigma_1 > \sigma_2|\tilde{r}) = 1$.
Which existing works assume/imply the latter assumption?
Furthermore, it raises the question of whether $P_{\Sigma_r}$ is indeed nonidentifiable if Theorem 3.2 has the same premise as Theorem 3.1.

4, there is no comparison with IRL methods (which often rely on the value function or the Q function instead of partial returns): intuitively how they are different, and empirically how they are different (e.g., by experimental results).

5, existing works (e.g., [9,10]), the reward function is define as a function of the state and the action or the state only (i.e., r(s,a), r(s)). Do these reward formulations affect the result in the paper?

6, there is only a toy grid world experiment (a very small state and action space, and linear reward functions) which is quite limited.

---

> ### Author Response · Authors · 2022-08-02
> **Response to reviewer, part 1**
>
> Thank you for your thoughtful review. We appreciate the opportunity to clarify certain technical details, the relationship of this work to inverse reinforcement learning, why noiseless preferences are informative, and why scalability is not our focus.
>
> **Would defining reward as r(s,a) or r(s) affect the results in the paper?**
>
> It would not. Our definition of reward is a generalization of reward defined over state or state-action pairs, which can be trivially mapped to reward over tuples of state, action, and next state. E.g., r(s,a) is mapped to an r(s,a,s') function simply ignoring the next state, s'.
>
> **"the proposed regret model is limited to the case of deterministic policies"**
>
> This is a misunderstanding. The regret model applies to stochastic policies as well. Our definition of a policy in L68-69 is in stochastic terms. Further, note that the algorithm relies on optimal advantage values; any state-action pair that has support (i.e., non-zero probability) in an optimal policy has an advantage of 0. That's true even if there is more than one action that is optimal from a state. And a policy that stochastically mixes multiple optimal actions from each state will also always create state-action pairs with an advantage of 0 (otherwise it would be sub-optimal). Additionally, please see our response to Reviewer ieTk under "What if there is a set of optimal policies…", which is relevant because any MDP with two optimal deterministic policies necessarily has infinite optimal stochastic policies (via mixing them as mentioned above).
>
> **"if I am not missing anything, Theorem 3.1 and 3.2 are not comparable... in Theorem 3.1, if $regret_{\tilde{r}}(\sigma_{1}) < regret_{\tilde{r}}(\sigma_{2})$, $P_{regret}(\sigma_1 \succ \sigma_2 | {\tilde{r}})>0.5$, in Theorem 3.2, if $\Sigma_{\tilde{r}}(\sigma_{1}) > \Sigma_{\tilde{r}}({\sigma_{2}})$, $P_{\Sigma_r}(\sigma_1 \succ \sigma_2 | {\tilde{r}})=1$."**
>
> Please see our response to all reviewers under "Noiseless preferences are not theoretically interesting or should be connected to past work". Building on that general response, note that they are very comparable in the sense that the setting that matches that of 3.2 is a special case of 3.1. In other words,  "if $regret_{\tilde{r}}(\sigma_{1}) < regret_{\tilde{r}}(\sigma_{2})$, $P_{regret}(\sigma_1 \succ \sigma_2 | {\tilde{r}})>0.5$" includes all cases where "if $regret_{\tilde{r}}(\sigma_{1}) < regret_{\tilde{r}}(\sigma_{2})$, $P_{regret}(\sigma_1 \succ \sigma_2 | {\tilde{r}})=1$". Therefore, 3.1 shows that regret is identifiable under the conditions of 3.2, if you swap partial return for regret and flip the > sign, as well as under other conditions.
>
> **"it raises the question of whether PΣr is indeed non-identifiable if Theorem 3.2 has the same premise as Theorem 3.1."**
>
> We agree and were careful to only claim that partial return is only non-identifiable in this noiseless setting. See L11 ("without preference noise..."), L47 ("without preference noise..."), L167 ("without the distribution over preferences providing information"). We suspect any scale-invariant regression model is identifiable with Boltzmann noise, so both models would be identifiable in that respect.
>
> **"the work needs to illustrate whether the proposed method is scalable to more practical problems, e.g., by including experiments with neural networks and larger state/action spaces"**
>
> We understand that some researchers value showing deep RL applicability of a new approach more than others. However, we don't claim scalability either as a contribution or a characteristic of regret-based preference models and openly discuss how it might be addressed in the latter portion of Appendix F.1. We consider scalability to be one of many important dimensions of engineering and experimentation, but it did not make the cut in this paper for reasons we detail in our general message to all reviewers, under "This paper does not demonstrate scalability to more complex environments."

---

> > ### Author Response · Authors · 2022-08-02
> > **Response to reviewer, part 2**
> >
> > **Please compare to inverse reinforcement learning (IRL) methods.**
> >
> > The inputs to IRL and learning reward functions from pairwise preferences (which appears to be the second most popular method of reward learning after IRL) are different. IRL requires demonstrations, not preferences over segment pairs. However, because a  a regret-based preference model always prefers optimal segments over suboptimal segments, at least one connection can be made. If one assumes that a demonstrated trajectory segment is noiselessly optimal (as in the foundational 2004 Abbeel and Ng IRL / apprenticeship learning paper), then such a demonstration is equivalent to expressing preference or indifference for the demonstrated segment over all other segments (or, equivalently, that no other segment is preferred over the demonstrated segment). However, IRL has its own identifiability issues in noiseless settings (see [http://proceedings.mlr.press/v139/kim21c.html](http://proceedings.mlr.press/v139/kim21c.html)) that, viewed from the lens of preferences, come in part from the "indifference" part of the above statement: since there can be multiple optimal actions from a single state, it's not generally correct to assume that a demonstration of one such action shows a preference over all others, and therefore it remains unclear in IRL what _other _actions are optimal. Note that since partial-return-based preferences can prefer suboptimal segments over optimal segments, the common assumption in IRL that demonstrations are optimal does not map as cleanly to partial-return-based preferences.
> >
> > As mentioned in our response to all reviewers under "Practically using regret is challenging", our regret preference model also relates to IRL in that many IRL algorithms require solving an MDP in the inner loop, like would be required for a perfect measure of regret while learning a reward function.
> >
> > An empirical comparison of IRL and learning reward functions from pairwise preferences would be valuable but is out of our scope—we are focused on an established technique that is not IRL, and its close relationship to IRL does not mandate that any research project test IRL algorithms too—and would be its own full research project. Further, how to do an apples-to-apples comparison is non-obvious, since the inputs are different.
> >
> > We have added a discussion of the relationship to IRL in App B.5.

---

> > > ### Comment · Reviewer_SeBN · 2022-08-08
> > > **Thank you for the response**
> > >
> > > I would like to thank the author for the response. It indeed resolves several concerns of mine.
> > >
> > > Unfortunately, my concern regarding the main theoretical contribution (Theorem 3.1 and 3.2) of the work remains.
> > > I am not convinced by the argument of the author that "3.1 shows that regret is identifiable under the conditions of 3.2 as the setting of 3.2 is a special case of 3.1". For example, as reviewer xps1 points out in his response, why does the first line of the proof of 3.1 hold if we change to the setting of 3.2 (can minimizing the cross-entropy even work in the setting of 3.2)? Maybe totally re-written proofs of the 2 theorems with a consistent setting are more persuasive. Furthermore, if the noiseless setting is not practical, the common setting should be the one from the logistic observation model in Equation 2 (the setting in 3.1).
> > >
> > > Besides, in my view, saying that this work does not focus on scalability to justify the experiment involves only a toy grid world environment is not convincing. While different in the inputs, the solution in this work and IRL have a similar computation bottleneck of an inner loop of policy optimization. However, works on IRL (e.g., the popular Guided Cost Learning: Deep Inverse Optimal Control via Policy Optimization by Chelsea Finn 2016) have impressive experimental results. This IRL work has been also cited in many works in the last 5 years.
> > >
> > > Therefore, I decide the keep my score.

---

> > > > ### Author Response · Authors · 2022-08-09
> > > > **Correcting our error**
> > > >
> > > > **We ask you to reconsider your score, given that you wrote that our original response _resolved several of your concerns_ and also depending on how well our responses below and above (in "Considering our contributions") satisfy your concerns about our lack of focus on scalability and our theoretical results.**
> > > >
> > > > **For @xps1 and @SeBN:** Thanks for further clarification about your concerns regarding the comparability of Thms 3.1 and 3.2. In our first response to your reviews, we focused in part on how the choice of optimization (max likelihood or otherwise) doesn't matter in the Thm 3.2 proofs, since noiselessly providing preferences based on partial return creates a many-to-one mapping.
> > > >
> > > > **Your point above about the proof of Thm 3.1 not applying to noiseless preferences is well taken, and we agree with you. Thank you for identifying this error.** We originally removed the temperature parameter from the softmax formula because it is redundant with the scaling of the reward and it simplified our notation to not include it. However, as you suggest, for the proof of Thm 3.1 to apply to the noiseless preferences setting of Thm 3.2, a revision is needed.
> > > >
> > > > Fortunately, **the required change is minor**. **We can add back the temperature as a optimizable parameter into Eqs 2 and 5 and allow a special case for temperature = 0, where the result is a hard max, i.e., if temperature = 0, the preference is given deterministically to the segment with the higher partial return in Eq 2 or regret in Eq 5.**
> > > >
> > > > With this tweak, when the temperature is 0, the proof of Thm 3.1 covers both the stochastic and noiseless settings, and the preference generator is realizable in the noiseless setting for Thm 3.1 and in the two proofs of Thm 3.2 (though these two proofs don't rely on the learning algorithm to provide their negative results). In other words, **Thm 3.1 does include the noiseless settings of Thm 3.2 as a special case, but both theorems' proofs required this "tweak" to have a learning algorithm that can realize the reward function that created the noiseless preferences.**
> > > >
> > > > The need for this added special case in the theoretical setting is now clear. However, in practical settings, the originally submitted algorithm (with no temperature) will simply scale reward until the improvement in loss after each epoch of learning is extremely small, which can be used as a stopping criterion, at which point the reward function results in nearly deterministic preferences. Such an approach is effective with noiseless preference labels in our experiments, as seen in Section 6.2 (with further detail in App F.2.1) and in the randomly generated _stochastic_ MDPs in App F.2.5.
> > > >
> > > > The revisions and clarifications above will be included in the final copy. In particular, we will revise to emphasize that _in noiseless preference label settings_, both theorems provide insight about whether the preferences contain the information required to recover the set of optimal policies (via directly recovering a reward function). **In other words, the theorems are really about each preference model _as a preference generator_, and the learning algorithm used in Thm 3.1 is merely meant to show _what information exists_ in each type of preferences.** For the negative result of Thm 3.2, its two proofs clearly show that some MDPs exist in which preferences based on partial return have the many-to-one mapping we referred to and therefore is not identifiable under _any_ learning algorithm. For the positive result of Thm 3.1, we merely need to show that there is one learning algorithm that permits identifiability in any MDP, which that proof does with the above tweak to allow a special hard max case.
> > > >
> > > > (Note that what we wrote above was summarized and rephrased in our response to @xps1 with the same subject, "Correcting our error".)

---

> > > > ### Author Response · Authors · 2022-08-09
> > > > **Considering our contributions**
> > > >
> > > > Regarding that we only use simple grid worlds (though up to 100 random versions of them), we can only make subjective arguments about the necessity of scalability in addition to our various contributions, which we understand may not persuade you, especially if your mind is already made.
> > > >
> > > > In our responses here, we have argued that the existence of deep IRL work—noting the inner-loop bottleneck—provides optimism that similar approaches can be applied to learning reward functions with regret-based preference models; without such optimism, the importance of demonstrating scalability in this paper would increase.
> > > >
> > > > Also, consider that **there are 100s of papers on IRL and there will only be _this one paper_ on reward learning from regret-based preferences, if you and our other reviewers decide it is worthy of publication. The first paper on IRL, in 2000 by Ng and Russell, also only focused on simple problems: two grid worlds and two versions of mountain car. We nonetheless share your strong interest in seeing regret-based reward learning scale to complex, real-world problems**.
> > > >
> > > > But we worry that it would be a disservice to progress in this popular form of learning reward functions to make showing scalability a hard constraint for publishing this _first_ paper, which we believe **provides urgently needed insight about the partial return preference model that continues to pervade new publications on learning reward functions from pairwise segment preferences, likely _including most that are currently in progress_ by other researchers.** If one instead considers whether the contributions we did make—a human study with empirical results, new preference model, theoretical results, synthetic preferences results, a new learning algorithm that approximates regret, and a clear existence proof that the previously overlooked preference model matters—are together sufficient for publication in NeurIPS, our contributions become a strength of this submission.

---

### Official Review · Reviewer_xps1 · 2022-07-11

**Rating:** 3
**Confidence:** 3
**Soundness:** 2 fair
**Presentation:** 1 poor
**Contribution:** 3 good

**Summary:**

This paper studies the problem of learning from human preferences. A new preference model is proposed which compares the advantage on the optimal policy between two trajectory segments. Some theoretical results are given that show the method is consistent under some assumptions and that a popular alternative is potentially not consistent. An algorithm is proposed to make use of this model. A new dataset is collected for a simulated delivery problem where workers provided preference labels for segments given to them. The proposed algorithm and model were compared with a partial return model on both a synthetic dataset and the collected dataset.

**Questions:**

- Can the authors clarify what assumption is being made over the covariates in the dataset $D_{\geq}$ in Definition 3.1?
- In Section 4, it mentions that data was collected via two different methodologies. In the end, was data from only the second used to present the results in the end or was it a mix?
- Can the authors provide more details about how the actual segments that were presented to the labelers were generated? In the appendix it just says that certain trajectories were favored, but it’s unclear to me what that means. Were these generated by demonstrations or an algorithm? Were they generated specifically to have good coverage over the state space?


**Limitations:**

See above discussion.

**Strengths And Weaknesses:**

Strengths
- The proposed preference model seems new to my knowledge and makes sense algorithmically.
- The theory could be interesting although I have some reservations about correctness and the significance of one of the results.
- The algorithm seems to perform significantly better than the partial return baseline in empirical evaluations.

Weaknesses
- The presentation could be improved. Generally speaking, there are many seemingly out of place paragraphs and sections that do not seem to advance the content of the paper. For example, Section 3 is about theoretical comparisons of the learning objectives but then Section 4 abruptly is about an experimental model, but we do not even know what the algorithm looks like or how we will use the models from Section 3 at this point. This is only explained in Section 6.
- The description of the algorithm is very confusing and way too informal to the point where I am still unclear on what it is actually being done.
    - What are the input rewards and policies? Why can they be empty sometimes?
    - Why would line 4 already estimate those? What procedure is being done to estimate these?
    - How many feature functions are necessary?
    - For the partial return algorithm, was the same framework used but the model just swapped out? How does this compare with past algorithms that use partial return? This seems important for the experimental comparison.
- The discussion of related work is quite sparse and it is difficult to place this work in the literature as a result.
- Theorem 3.1 (and Definition 3.1) is dubious. Implicitly there is an assumption that the dataset covers every $\sigma_1$ and $\sigma_2$ pair and infinite data for each one. Not just that it contains infinite data for some distribution over certain segments, which is what Definition 3.1 is ambiguous about. This only becomes apparent in the proof where this unstated fact is used.
- I am also not convinced that Theorem 3.2 is all that significant. The setting is different from the preliminaries of the paper: to assume noiseless labels is to say that $P(\cdot | r)$ is actually not in the class of models from Section 2.2 since the softmax cannot realize this model for finite rewards. The problem is therefore misspecified, and it’s not surprising that there is an identifiability issue, considering that the KL divergence cannot even be driven to zero if the true reward function is plugged in! Note that Theorem 3.1 crucially made use of the fact that the model is realizable to prove the positive result, so I feel that this is an unfair comparison.
- The computational burden of solving potentially many MDPs to optimize the proposed preference model seems difficult to overcome.

---

> ### Author Response · Authors · 2022-08-02
> **Response to reviewer, part 1**
>
> Thank you for your suggestions on improving presentation, including for theoretical clarification. We heartily welcome suggestions for improvement and have addressed many of yours in the uploaded diff in the supplementary material.
>
> **"In Section 4, it mentions that data was collected via two different methodologies. In the end, was data from only the second [collection methodology] used to present the results in the end or was it a mix?"**
>
> Both stages' data was used unless otherwise stated. We have added this missing information in Sec 4.3.
>
> **"... more details about how the actual segments that were presented to the labelers were generated?"**
>
> Yes! We agree such detail is needed and have added it (see App D.3 in the uploaded diff).
>
> **The section organization is confusing. I did not see an algorithm until Sec 6.**
>
> With your feedback in mind, we have revised the paper to be much clearer regarding a single point that should help with your confusion. Specifically, Section 2 defines complete algorithms through Equations 1 and 2 for partial return and Equations 1 and 5 for regret. This algorithm assumes that regret can be exactly measured, and the Sec 3 theory makes this same assumption. Sec 4–6 comprise our experimental analyses (not counting what is in the appendix), with Sec 4 about data gathering, Sec 5 about results _without_ reward function learning, and Sec 6 about results from reward function learning. Alg 1 is presented in Sec 6 because it focuses on creating approximations of regret for its practical application in our experiments, and otherwise it is the same algorithm as in Section 2, which is packed into line 10 of the Alg 1. The uploaded diff shows our revisions to be clearer: Sec 2.3 has a new paragraph entitled "Algorithms in this paper" that explicitly states that two algorithms have been defined (with clarification on what they are), points forward to the algorithms that will be defined, and connects the 3 main algorithms with which results they are used to obtain; and the first sentence of Sec 6.1 now clearly states that it is only introducing an approximation of regret to be used in the algorithm from Sec 2.
>
> **For the partial return algorithm, was the same framework used but the model just swapped out? How does this compare with past algorithms that use partial return?**
>
> Alg 1 is not used for learning reward functions with the partial return preference model, since no approximation of regret is needed for learning via partial return. All reward learning with a partial return preference model uses the algorithm from Sec 2. Our algorithm using partial return matches that of numerous past works, including the most cited one on this topic, by Christiano et al.
>
> **"... discussion of related work is quite sparse" / "difficult to place this work in the literature as a result"**
>
> Instead of an explicit related works section, we discuss related work throughout the paper, where each work is related to the immediate content of the paper. Putting all of these discussions with citations together adds to a substantial discussion. As far as placing the work in the literature, this work is a direct response to the unexamined and taken-on-faith partial-return assumptions of existing literature ([9–16]. We welcome any suggestions on how we might make that clearer than we do in L25-33, and any suggestions for additional work we should discuss.

---

> > ### Author Response · Authors · 2022-08-02
> > **Response to reviewer, part 2**
> >
> > **"The description of the algorithm is very confusing"** // **"What are the input rewards and policies? Why can they be empty sometimes? Why would line 4 already estimate those? What procedure is being done to estimate these? How many feature functions are necessary?"**
> >
> > We trust that the above explanations help somewhat with your confusion regarding Algorithm 1. Further, without familiarity with doing general policy iteration (GPI) with successor features (SFs) (e.g., via reading Barreto et al. [24]), the description of Algorithm 1 will necessarily seem very complex, but it's a straightforward application of this concept to approximate Q and state values for arbitrary reward functions. Successor features are a potent tool with applications across RL.
> >
> > Regarding your specific questions, SF functions are learned _per policy_. So an input set of policies gives us a set of SF functions. We can also use a set of reward functions to create or augment this set of policies, by adding an approximately optimal policy for each reward function. To end up with a set of policies to create SF functions with, we can therefore use either or both of an input set of policies and reward functions. Crucially, these reward functions are _not_ the learned reward function that is the algorithm's goal. They merely are a step in a process that allows us to approximate regret of a segment for an arbitrary reward function. What reward functions and policies should be inserted is an important open question for SF-based methods in general, but our sense is that the performance of GPI under SF-based methods is improved with a greater diversity of SF functions (via a diverse set of policies) and by having some policies that perform decently (but not necessarily perfectly) on reward functions for which state and Q values are being estimated via GPI.
> >
> > To determine what number of feature components should be used is essentially asking what features are needed to linearly model a reward function. If the reward function is linear and its components/features are known, then only those features should be used. At the other extreme, if the state and action space are discrete, one could know nothing about the reward function and yet linearly model all possible reward functions by creating a feature for each (s,a,s') that is 1 for (s,a,s') and 0 otherwise. If either are continuous and the reward function's linear features are unknown or the reward is nonlinear, then clear guidance requires the follow-up work on scalability that we discuss elsewhere in these comments as out of scope and also near the end of App F.1.
> >
> > **"Theorem 3.1 (and Definition 3.1) is dubious. Implicitly there is an assumption that the dataset covers every σ1 and σ2 pair and infinite data for each one." // "what assumption … over the covariates"**
> >
> > You are right that we made this assumption, but it was explicit. We explicitly stated this assumption in L137-138: "Further assume that in this dataset, all possible n-length segment pairs appear infinitely many times." This type of assumption is standard in identifiability proofs, where one wants to know if the parameters of a data-generating model can be recovered under the highly favorable conditions of infinite, exhaustive data. Additionally, no assumption is being made over the covariates; as long as all segment pairs appear infinitely in the data, the proof holds.
> >
> > **"P(⋅|r) is actually not in the class of models from Section 2.2… The problem is therefore misspecified, and it’s not surprising that there is an identifiability issue… an unfair comparison."**
> >
> > As you correctly point out, Thm 3.1 shows that the class of models from Sec 2.2 achieves identifiability with regret-based preferences. However, Thm 3.2, as a negative result, does need to rely specifically on the class of models from Sec. 2.2. Rather, Thm 3.2 shows that there are MDP\r tasks in which _there is no class of models that _can recover an equivalent reward function from partial-return-based preferences if the preference generator noiselessly prefers according to partial return. Specifically, we show that the mapping from two reward functions with different sets of optimal policies to partial-return based preferences is a many-to-one-mapping, and therefore the information simply does not exist to invert that mapping and recover a reward function with the same set of optimal policies. We state this more clearly in the uploaded diff.

---

> > > ### Comment · Reviewer_xps1 · 2022-08-08
> > > **Thanks for the response**
> > >
> > > Thanks for your response. I appreciate the details and clarifications.
> > >
> > > > Noiseless preferences are not theoretically interesting or should be connected to past work. (@xps1, @SeBN)
> > >
> > > I’m not sure I understand this. My concern with this section was not related to the theoretical significance of noiseless settings (by the way, I do think it’s important). I don’t think SeBN suggested this either, so can you clarify what you mean by this?
> > >
> > >
> > > In any case, I don’t think either of our points have been addressed here. I'm skeptical of the claimed difference between the regret model and the return model that is presented in Section 3 because it does not appear to be a fair comparison. As SeBN put it quite well: 3.1 and 3.2 do not satisfy the same premise. The noiseless setting falls outside the problem setting established in Eq 2 and 5 where a logistic model is used - it’s misspecified for this model class for finite rewards. Meanwhile, Theorem 3.1 is using this realizability to derive the positive result. This is used in the first line of the proof. I think the comparison would be fair if either the negative result of Theorem 3.2 were strengthened to the case where preference function is realizable but it still fails to be identifiable, or the positive result of Theorem 3.1 were shown to extend to a non-trivial class of problems where there is misspecification.

---

> > > > ### Author Response · Authors · 2022-08-09
> > > > **Correcting our error**
> > > >
> > > > > I’m not sure I understand this. My concern with this section was not related to the theoretical significance of noiseless settings (by the way, I do think it’s important). I don’t think SeBN suggested this either, so can you clarify what you mean by this?
> > > >
> > > > **We misinterpreted your statement** "... not convinced that Theorem 3.2 is all that significant" to be about the noiseless setting, but, assuming we now understand you better, your issue was more specifically with the comparability of 3.1 and 3.2 and that a reward function providing noiseless partial-return-based preferences is not realizable under the algorithm we use in Sec 2 and Definition 1 (and is the common algorithmic component of [9-16]). **We appreciate your added clarification and that your issues do not regard the importance of the setting of Thm 3.2 but rather some technical aspects, since that means we agree that the theoretical problem is impactful and can focus on the solution.**
> > > >
> > > > > In any case, I don’t think either of our points have been addressed here. I'm skeptical of the claimed difference between the regret model and the return model that is presented in Section 3 because it does not appear to be a fair comparison. As SeBN put it quite well: 3.1 and 3.2 do not satisfy the same premise. The noiseless setting falls outside the problem setting established in Eq 2 and 5 where a logistic model is used - it’s misspecified for this model class for finite rewards. Meanwhile, Theorem 3.1 is using this realizability to derive the positive result. This is used in the first line of the proof. I think the comparison would be fair if either the negative result of Theorem 3.2 were strengthened to the case where preference function is realizable but it still fails to be identifiable, or the positive result of Theorem 3.1 were shown to extend to a non-trivial class of problems where there is misspecification.
> > > >
> > > > **We agree with your points above and thank you and @SeBN for pushing back to reveal this error.** In our response to you and @SeBN that is in @SeBN's thread, we share **a small revision to solve the realizability issue** for noiseless preferences. The issue you two identified actually affects both Thm 3.1 (in the noiseless subset of its settings) and Thm 3.2, so the fix for realizability should be applied to both. We also want to emphasize that in our final copy, we will clarify that **the theorems focus on what information such infinite datasets hold when either preference model acts as the preference generator, where any usage of a learning algorithm in the proof Thm 3.1 only serves to show that the information exists in the dataset to recover an equivalent reward function**.

---

### Official Review · Reviewer_dwHc · 2022-07-11

**Rating:** 4
**Confidence:** 2
**Soundness:** 2 fair
**Presentation:** 3 good
**Contribution:** 2 fair

**Summary:**

The paper studies the issues of reward functions. It considers alignment on the pairs of trajectory segments from human-generated preferences. The paper finds that it is flawed that human preferences are assumed to be informed by partial return. Previous works consider the sum of rewards over a segment. The paper provides an alternative preference model based on the regret of each segment.


**Questions:**

What is the segment’s desirability?
Figure 1 is a little confusing.  In Figure 1, the right segment should have a higher sum of reward according to humans’ preference.
How to define a start state of a segment? And how to define an end state of a segment?

Regret is computed based on a segment. The segment is also partial.
What is the advantage of using regret?
The motivation of using desirability is not clear.


**Ethics Review Area:**

["I don’t know"]

**Limitations:**

It would be better to provide more results and more details about experiments.
The paper needs to compare with other baselines, such as methods using (2).

**Strengths And Weaknesses:**

The paper addresses an interesting problem about human preferences. It shows that the partial return preference model can prefer suboptimal actions with lucky outcomes.
The paper also provides a method based on the regret of each segment, which is equivalent to the negated sum of an optimal policy’s advantage of each transition in the segment.
The paper also provides theoretical comparisons.

The main weakness is that the experimental results seem not to be enough. The proposed method does not provide comparison with previous methods from [9-16].
The environments are also limited.
It would be better to provide more results.
Another weakness is that the code is not opened. It would be difficult to reproduce the results by others.

---

> ### Author Response · Authors · 2022-08-02
> **Response to reviewer**
>
> Thank you for your suggestions and in particular for your encouragement to open our code and data.
>
> **The code should be open sourced.**
>
> When we submitted, we wrote in the NeurIPS submission checklist (L478–481) that we were seeking approval for open sourcing the code and data, which includes the learning code, the Mechanical Turk UI for gathering human preference data, and our human preferences dataset. Good news: it was approved and all will be shared! We could not find any previous paper that shared a human dataset (and many never test their algorithm for "human" preferences with real humans), so we are excited to be providing the first such dataset for others to reproduce our work and do novel research with the dataset.
>
> **"compare with other baselines, such as methods using [partial return]"**
>
> The commonality of [9-16] is that they use partial return and (usually) likelihood maximization, which is _precisely_ the baseline that we use throughout our paper. Future work could compare using different preference models with each of these papers' novelties, such as choosing segment pairs for max info gain, but such work would be extending the baseline we already focus on to new settings, which would be informative but outside our scope. We do want to draw your attention though to the 4 additional baselines we introduce in Appendix B, each receiving more limited evaluation than our partial-return baseline. Since submission, we did however add in App B.3 and F.3 a second baseline, an "expected return" preference model, which is halfway between the partial return and deterministic regret models, considering each segment's partial return and end state value (but not start state value).
>
> **"What is the segment’s desirability?"**
>
> Desirability is not a technical word in this paper. We only use it in a colloquial sense to mean "characteristics that would make something (e.g., segment or a reward) more preferable than something else". We will clarify in the final version.
>
> **"environments are also limited" / "more results"**
>
> Please read why we focus on simple environments for scientific reasons, under "This paper does not demonstrate scalability to more complex environments" in our general comment to our reviewers. We also note that though our experiments were limited to grid worlds, our synthetic preferences experiments include learning and testing reward functions 100 different randomly generated grid world MDPs. Regarding the amount of results, we have theoretical results; 6 sets of results with synthetic data: the main one in Section 6.2 and 5 more in Appendix F.2; and a large human subjects experiment tested—which required months of UI design and iterative tuning—with correlational and likelihood results in Section 5 and reward-learning results with 3 preference models (Sec 6, but see App F.3 for all 3 models). As described under "Updates" in our comment to all reviewers, a new baseline has been added since submission, in the uploaded diff's App B.3 and F.3.
>
> **"more details about experiments" / "How to define a start state of a segment… [and] end state of a segment?"**
>
> For space considerations we put most experimental detail in our appendix. We have added more detail about how segment pairs were chosen for human labeling (App D.3), which implicitly includes how start and end states are chosen. We will also open source our code, which will provide a secondary form of experimental detail.

---

> > ### Comment · Reviewer_dwHc · 2022-08-07
> > **Thanks and some clarification**
> >
> > I thank the authors for their responses. However, I notice that some of my comments were not completely addressed.
> >
> > Do you mean partial turn indicates [9-16] in the experiments? Why just one implementation?
> > Do you need to compare with some safety rl methods, like CPO?
> >
> > Figure 1 does not seem to help improve understanding of the idea. It would be better to provide more explanation about Figure 1 or change an example. Why not add some related work about safety RL?

---

> > > ### Author Response · Authors · 2022-08-08
> > > **To improve our clarity further, some explanations and questions**
> > >
> > > We appreciate you following up on these questions and topics. We're happy to discuss further and hope to fully understand your confusion and preempt similar confusion for future readers through revisions for clarity.
> > >
> > > Regarding **safe RL**, this paper focuses on the problem of how to _learn a reward function_ that is aligned with the interests of human stakeholders, specifically focused on the well-established technique of learning from preferences over pairs of trajectory segments. To our knowledge, safe RL such as CPO, on the other hand, generally assumes that a reward function and a set of constraints _are already given_ and focuses on RL algorithms that do policy improvement while making various guarantees to avoid violating the safety constraints. In short, reward learning and safe RL are different problems that do not seem particularly related. We do see how a different usage of the word "safety" could encompass the alignment of reward functions—since poorly aligned objective functions generally are a significant societal threat—despite the safe RL literature not focusing on such alignment.
> > >
> > > We do not understand your question about partial return and [9-16]. Could you elaborate on what you mean and why more than one implementation would be desirable?
> > >
> > > Regarding **Figure 1**, we will add some more explanation below to see if any of it addresses your confusion. We implicitly assume a reward function that—in these two segments—only deviates from 0 to reflect damage to the vehicle or harm to the passengers. If it would help to make that information more explicit, please let us know and we will add it. The left segment involves no damage to the vehicle and therefore has 0 reward, even though it leads the car to an end state in which the car will almost certainly have a horrible collision within seconds. The right segment involves minor damage but otherwise does roughly the reverse of the left segment: it takes the car from a start state in which collision will likely happen soon to a safe end state. (Additionally, are you confused about how the figure is communicating the start states and end states, which are simply the beginning and end of the arrows / where the two images of the car are? If so, we can also write that information more clearly.) So by looking at partial return alone (i.e., the sum of reward along the segment but _not_ the start or end state values), the left segment is better despite being suboptimal, but human intuition prefers the right (optimal) segment, and the right segment has a lower deterministic regret (Eq. 3) because deterministic regret does include the start and end state values.

---

> > > > ### Comment · Reviewer_dwHc · 2022-08-08
> > > > **Feedback again**
> > > >
> > > > >Regarding safe RL, this paper focuses on
> > > >
> > > > Thank you. Figure 1 seems to be a safe RL example. That is why I’m a little confused about the motivation.
> > > >
> > > > > about partial return and [9-16]
> > > >
> > > > Sorry for the unclear part. In the experiments, there are baselines named as Partial return (noiseless) and Partial return (stochastic). Are the two baselines from [9-16] or created by the paper? There are a lot of partial return methods. Why not compare with the methods from those papers?

---

> > > > > ### Author Response · Authors · 2022-08-09
> > > > > **Resolving remaining clarification questions**
> > > > >
> > > > > > Thank you. Figure 1 seems to be a safe RL example. That is why I’m a little confused about the motivation.
> > > > >
> > > > > Thank you for explaining your confusion. Although Figure 1 may indeed remind readers of research on safe RL (and safety in autonomous driving, which employs a different definition of the word "safe"), here Figure 1 is used as a simple example of how human preferences intuitively do not seem to given based on the partial return of a trajectory (or else the left trajectory would be preferred). We hope that this exchange has helped clarify that this work is unrelated to safe RL; it solely focuses on more accurate inference of the reward functions that drive human preferences. Any RL algorithm can then be used to learn a task from such a human-aligned reward function. If an RL practitioner decided to add safety constraints to such a learned reward function (which however is certainly not the focus of this paper), they could then use various safe RL algorithms to learn from the reward function—created through regret preference models—such as CPO.
> > > > >
> > > > > > Sorry for the unclear part. In the experiments, there are baselines named as Partial return (noiseless) and Partial return (stochastic). Are the two baselines from [9-16] or created by the paper? There are a lot of partial return methods. Why not compare with the methods from those papers?
> > > > >
> > > > > Understood. Thanks for communicating back and forth with us to get to this point of clarification! **The partial return (stochastic) preference model is taken exactly from [9-16] and is common to all of them. And the "partial return (stochastic)" reward learning algorithm—created from joining both the preference model (Eq 2) and maximizing likelihood (Eq 2)—is the common part of each learning algorithm from [9-16].** L96-97 of the currently downloadable version (lines unchanged from the original submission) were meant to state this relationship. We will be more explicit in the final version to prevent confusion like yours. Comparing preference models, which was not the focus of _their_ papers, is our focus. Instead, they focused on largely orthogonal algorithmic elements, like what segment pairs to present for elicitation [13] or how to maximize likelihood (Eq 1) with a deep neural network representation of the reward function [9]. So this baseline appears to be precisely what you are asking for.
> > > > >
> > > > > The partial return (noiseless) preference model is _only_ used to generate preferences, not for the reward learning algorithm, as we explain at the start of Section 6.2 (L327–331 in the currently downloadable version). This noiseless preference generator is not typically used by [9-16]. You can see in our comments to all reviewers an explanation of the importance of these noiseless versions of each preference model, despite that they are not typically employed in related work.

---

### Official Review · Reviewer_ieTk · 2022-07-13

**Rating:** 7
**Confidence:** 4
**Soundness:** 3 good
**Presentation:** 3 good
**Contribution:** 3 good

**Summary:**

The paper deals with the problem of modeling human preferences in a setting where they are presented with two segments and choose to compare the two. Importantly, the authors propose that people compare the segments in terms of their values (measured by a notion of regret) instead of the total reward achieved by each segment. A regret notion---the sum of the negative advantage function value of the segment---is used to summarize the value of a segment. The advantage function is defined under an optimal policy pi^* (optimal wrt the true reward r) and the reward of interest \tilde{r}.

**Questions:**

Could the authors elaborate more on the setting when there are multiple optimal policies, how one should define regret in such cases, and how the results in Section 3 look like when one chooses different optimal policies to define ther regret?


-----

L63: typo in the mathematical expression

**Limitations:**

Yes, the authors have discussed the limitations and societal impact of their work in Appendix A. (When writing the above reviews, I have not read Appendix A of their paper yet.)

**Strengths And Weaknesses:**

Strength: The paper provides a key insight that preferences are not just a function of the (instantaneous) reward of the segment of interest. Given that humans may care about the goal of a task, their preference towards different segments will necessarily be influenced by their long-term value, i.e., how these segments perform in terms of achieving the goal. The identification results presented in Section 3 nicely summarized the proposed insight.


Weakness: The biggest weakness to me is how one can utilize this insight in practice. In particular, calculating the regret requires one to know the optimal policy under the true reward r (or at least the value and q-value function of that optimal policy under any reward \tilde{r}). In general, even if one provides an optimal policy \pi^*, the planning problem (and possibly learning problem when the transition matrix is unknown) will make the estimation of the regret very hard. As a related note, in the general framework, the regret definition under a particular optimal policy and L150 ("And regret(\sigma|r) > 0 if and only if \sigma is suboptimal") in the proof seem to only hold if one assumes there is one single optimal policy \pi^*. What if there is a set of optimal policies under the true reward r? How should one adjust the regret notion for that?

---

> ### Author Response · Authors · 2022-08-02
> **Response to reviewer**
>
> We thank you for your kind words and for communicating your clear understanding of the paper.
>
> **"What if there is a set of optimal policies…"?**
>
> Neither our core algorithm (Section 2) nor our approximation algorithm (Alg 1) assume a single policy. Note that regret is defined over optimal advantages, and all by definition optimal policies have the same Q values and state values (e.g., see Sutton and Barto)—and therefore the same advantages—from any state-action pair, so our definitions are not affected by having more than one π*.
>
> **"biggest weakness … calculating the regret requires one to know the optimal policy under the true reward r (or at least the value and q-value function of that optimal policy under any reward \tilde{r})"**
>
> This is an easy misunderstanding to make. The algorithm requires knowing the optimal state and Q values—or some approximation of them—for the current candidate reward function, which is _not _dependent on the optimal policy for the true reward function. This solve-an-MDP-in-the-inner-loop problem is common to many IRL algorithms, including in the foundational 2000 IRL paper by Ng and Russell, where numerous approximations have been developed to handle the problem tractably. Our Alg 1 addresses this issue in certain settings, and the latter portion of App F.1 discusses how to do so in deep RL settings, as Brown et al. have done already for learning reward from preferences.
>
> **L63 typo**
>
> Good catch and thank you! Fixed.

---

### Author Response · Authors · 2022-08-02
**Response to all reviewers, part 1**

We thank our reviewers for their time and thoughtful consideration. We are encouraged that our reviewers wrote that our submission focuses on an interesting problem (@dwHc), provides a "key insight" that human preferences are based on more than just instantaneous reward of segments (@ieTk), contains a "sound and reasonable" derivation of the segment's regret as the advantage function (@SeBN), and shows that our novel regret-based preference model significantly outperforms the pre-existing partial-return-based model in experiments (@xps1). In particular, we are grateful that your reviews have led to what we consider to be vast improvement in our submission, viewable via the new version we have uploaded and in **the diff we included in supplementary materials**.

**[Recap] What is our goal?** A _scientific_ study of the effect of using different preference models when learning reward functions from human preferences, which had previously been unexamined.

**[Recap] Why this goal?** To correct a perceived mistake in recent research—assuming human preferences are unaffected by state values of trajectory segments—and to create a strong foundation for further research based on insights from theory and simple domains with minimal approximation error.

**[Recap] How do we achieve our goal?** We reveal flaws in the commonly assumed partial-return preference model; propose an improved preference model (based on regret); compare them theoretically and with real human data and synthetic data; and show that the choice of preference model can be impactful.

**[Recap] What is not the goal?** To demonstrate scalability to complex problems, which is important but beyond scope. Such a focus would force coarser approximations of regret, which would obscure the scientific analysis this paper focuses upon. We elaborate in the scalability part of "Common concerns" below.

**-Updates-**

**Open source and data.** (@dwHc) As foreshadowed in our NeurIPS checklist, we are thrilled to share that we have approval to **open the learning and experimentation code, the software for the UI and backend to gather human data on Mechanical Turk, and our human preferences dataset.** We could not find any previous paper that had shared a dataset of preferences from real humans and are particularly excited that we can make the first such dataset available.

We have also added **another baseline preference model**, based on expected return of a segment, and tested it on the human preference dataset (in App B.3 and App F.3 of the uploaded diff in the supplementary material). The regret preference model matches or exceeds the performance of this model as well.

---

> ### Author Response · Authors · 2022-08-02
> **Response to all reviewers, part 2**
>
> **-Common concerns among our reviewers-**
>
> **Practically using regret is challenging.** (@ieTk, @xps1, @SeBN)
>
> We agree yet do not find this assertion problematic. We have provided evidence—theoretically and with experimentation—that regret is more effective when precisely measured or effectively approximated. The challenge of efficiently creating such approximations presents one clear path for future research and is not a good argument for staying within the local maximum of partial-return-based preference models.
>
> In fact, like our regret model, inverse reinforcement learning (IRL) was founded on an algorithm that requires solving an MDP in an inner loop of learning a reward function. (For example, see the foundational 2000 Ng and Russell paper on IRL and also Algorithm 1 in the recent IRL survey at [https://arxiv.org/pdf/1806.06877.pdf](https://arxiv.org/pdf/1806.06877.pdf).) This challenge hasn't stopped IRL from being an impactful problem, and handling this inner-loop computational demand is the focus of much IRL research.
>
> Appendix A.3 has been added to address this topic.
>
> **This paper does not demonstrate scalability to more complex environments.** (@ieTk, @SeBN)
>
> We agree. The focus of this paper is not deep RL or complex tasks, so such scalability—while highly important—was not in scope. As we mention above, this paper seeks a _scientific_ analysis of human preferences and preference models, which conflicts with scaling up to tasks commonly used for deep RL. Specifically, testing in more complex tasks would require coarser approximations of regret, which, as stated earlier, can make results harder to interpret. For example, if a regret-based algorithm fails, we wouldn't know whether regret is to blame or whether the approximation of regret is to blame.
>
> Further, we believe that algorithms for human-created data should be tested with _human_-created data, only using synthetic data to better understand the algorithms. Gathering human datasets is highly expensive in terms of design and engineering (see [https://bit.ly/humanprefs](https://bit.ly/humanprefs) for our UI that trains subjects and elicits their preferences). Testing these models in other tasks with real human data would have required multiple costly human subjects studies. When the cost of a human subjects study is properly appreciated in contrast to that of a purely computational experiment, we trust that our reviewers will agree that our combined 3 proofs, 3 evaluations of our human preferences dataset (Secs 4 and 6.3, extended in App F.3), and 6 synthetic-data evaluations (Sec 6.2 and App F.2) are more than sufficient for a single paper. Our contribution of a released human dataset will make testing in multiple environments easier for future research efforts in this area.
>
> In future work on the _application_ of regret preference models, we or other researchers can face the _engineering_ task of scalability. Given that IRL has made tremendous progress in this direction and the cited Brown et al. [30] paper has scaled an algorithm with similar needs to those of Alg 1, we are highly optimistic that the methods to scale can be developed and probably already exist (e.g., in [30] and in the later part of our App F.1, under "Instantiating Algorithm 1 for reward functions that may be non-linear").
>
> **Noiseless preferences are not theoretically interesting or should be connected to past work.** (@xps1, @SeBN)
>
> We explain below (and at the end of Appendix C in the uploaded diff in the supplementary material) why noiseless preferences are important.
>
>
>
> 1. An intuitive argument: Noise is often motivated as modeling human error. Having an algorithm rely on noise—structured in a very specific, Boltzmann-rational way—is an undesirable crutch. Noiseless preferences are also feasible, if rare.
> 2. In many related settings, noiseless data is assumed for theory or derivations. For instance, the foundational IRL paper by Abbeel and Ng on apprenticeship learning treats demonstrations as noiselessly optimal. Also, "Reward identification in inverse reinforcement learning", an ICML 2021 paper, focuses on identifiability with noiseless, perfect demonstrations.
> 3. Having structured noise provides information that can help both preference models, but these proofs show that there are cases where the signal behind the noise—either regret or partial return—isn't enough in the partial return case to identify an equivalent reward function. So, in a rough sense, regret is better at using both the signal and the noise, which might explain its superior sample efficiency in our experiments, across both human labels and synthetic labels. Relatedly, the noiseless setting can help us understand each preference model's sample efficiency in a low-noise setting.

---

### Meta-Review · Area_Chair_6fyf · 2022-08-24

**Recommendation:** Reject
**Confidence:** Less certain

**Metareview:**

The submitted paper was reviewed by 4 knowledgable reviewers and the reviewers and authors enganged in intense discussions. The authors clarified many details in these discussion but could not convince the reviewers in all regards (there are still open concerns regardings the proofs and the update proofs came in rather late so that there was insufficient time for the reviewers to further interact; there concerns regarding experiments although I discounted most of those regarding to scalability as I agree with the authors in that regard to some extent; etc.). Moreover, looking at the discussions and the authors' responses, the paper would benefit from making several points more clear/improving their presentation, likely by including parts which came up in the discussions in the paper. Considering all this, I think this paper should go through another round of reviews before it should be accepted and I am recommending rejection of the paper. Please note that it was not easy to come to this decision - there are some important insights and experiments in the paper which should be made available to the community asap. Thus I would honestly encourage the authors to improve their paper considering the reviewers' comments and take-aways from the discussion and submit a revised version of the paper at one of the upcoming conferences. I am already looking forward to seeing an improved version of the paper being published.

**Award:**

No

---

### Decision · Program_Chairs · 2022-09-14

Reject